# Trial protocol: a multicentre randomised trial of first-line treatment pathways for newly diagnosed immune thrombocytopenia: standard steroid treatment versus combined steroid and mycophenolate. The FLIGHT trial

Julie Pell,[1] Rosemary Greenwood,[2] Jenny Ingram,[2] Katherine Wale,[3] Ian Thomas,[1] Rebecca Kandiyali,[3] Andrew Mumford,[4,5] Andrew Dick,[4,6] Catherine Bagot,[7] Nichola Cooper,[8] Quentin Hill,[9] Charlotte Ann Bradbury[4,5]

For numbered affiliations see end of article.

**Correspondence to**
Dr Charlotte Ann Bradbury;
c.bradbury@bristol.ac.uk

## ABSTRACT

**Introduction** Immune thrombocytopenia (ITP) is an autoimmune condition that may cause thrombocytopenia-related bleeding. Current first-line ITP treatment is with high-dose corticosteroids but frequent side effects, heterogeneous responses and high relapse rates are significant problems with only 20% remaining in sustained remission with this approach. Mycophenolate mofetil (MMF) is often used as the next treatment with efficacy in 50%–80% of patients and good tolerability but can take up to 2 months to work.

**Objective** To test the hypothesis that MMF combined with corticosteroid is a more effective first-line treatment for immune thrombocytopenia (ITP) than current standard of corticosteroid alone.

**Methods and analysis** Multicentre, UK-based, open-label, randomised controlled trial.

**Setting** Haematology departments in secondary care.

**Participants** We plan to recruit 120 patients >16 years old with a diagnosis of ITP and a platelet count <30x10⁹/L who require first-line treatment. Patients will be followed up for a minimum of 12 months following randomisation.

**Primary outcome** Time from randomisation to treatment failure defined as platelets <30x10⁹/L and a need for second-line treatment.

**Secondary outcomes** Side effects, bleeding events, remission rates, time to relapse, time to next therapy, cumulative corticosteroid dose, rescue therapy, splenectomy, socioeconomic costs, patient-reported outcomes (quality of life, fatigue, impact of bleeding, care costs).

**Analysis** The sample size of 120 achieves a 91.5% power to detect a doubling of the median time to treatment failure from 5 to 10 months. This will be expressed as an HR with 95% CI, median time to event if more than 50% have had an event and illustrated with Kaplan-Meier curves. Cost-effectiveness will be based on the first 12 months from diagnosis.

**Ethics and dissemination** Ethical approval from NRES Committee South West (IRAS number 225959). EudraCT

### Strengths and limitations of this study

► First UK multicentre randomised controlled trial for first-line treatment of immune thrombocytopenia.
► Tests a pragmatic, cost-effective approach which if effective, may be applicable to other autoimmune conditions.
► The trial includes patient-oriented outcomes by using validated questionnaires to assess quality of life, fatigue, impact of bleeding and care costs.
► Option to consent to additional blood samples for translational research to maximise scientific potential.
► The limitations include the open-label design, lack of very long-term follow-up and sample size unable to detect small differences between treatment arms.

Number: 2017-001171-23. Results will be submitted for publication in peer-reviewed journals.
**Trial registration number** NCT03156452

## INTRODUCTION

Immune thrombocytopenia (ITP) has an incidence of 2.9/100 000 person-years.[1] It is an autoimmune condition that may present with bleeding and bruising due to a low platelet count. In ITP, there is increased consumption and reduced production of platelets due to both antibody and cell-mediated autoimmune attack of platelets and megakaryocytes involving dysregulated autoreactive T and B cell lymphocytes.[2–5]

ITP can be classified according to the duration of illness into newly diagnosed (<3 months), persistent (3–12 months) and chronic (>12 months).[6] ITP may also be

classified as either primary when it presents in isolation or secondary when ITP occurs in the context of an associated illness or medication.[6]

ITP is a diagnosis of exclusion and made when the platelet count <100×10$^9$/L and other causes of thrombocytopenia are excluded by history, examination and laboratory evaluation.[6 7]

Current first-line ITP treatment is with high-dose corticosteroids but this has several downsides. First, the majority of patients suffer significant side effects including mood swings, difficulty sleeping, weight gain, high blood pressure, diabetes, gastric irritation, skin thinning and osteoporosis. A published survey of patients with ITP reported 98% had at least one side effect and 38% stopped or reduced dosage due to intolerable side effects.[8] In the UK ITP registry, the most frequently reported comorbidities were related to corticosteroids and correlated with duration of treatment (hypertension in 30%, diabetes in 19%).[9] The second problem is that patients are heterogeneous in their response to corticosteroid with some (approximately 20%) not responding at all and the majority of others (70%–90%) relapsing when the corticosteroids are reduced or stopped.[7 10 11] Patients, who are refractory or relapse (the majority), remain at risk of bleeding/bruising, which occasionally can be severe including intracranial haemorrhage.[12] They often receive more corticosteroid with associated side effects. Some require hospital admission and expensive rescue therapies (eg, intravenous immunoglobulin, Ig for a 70 kg patient=£3906). They continue to require frequent blood tests and doctor visits and are usually unable to continue their normal activities until their illness is controlled. Fatigue is also associated with disease activity and can be severe.[13] Physical factors combine with psychological stress through fear of bleeding, need for time off work and lifestyle restrictions due to bleeding risk to adversely impact quality of life.[14 15]

First-line treatment for ITP is unsatisfactory but it remains unchanged for decades. Although a small number of studies have tested alternative approaches, a well-tolerated, effective and durable new approach has not been conclusively demonstrated. High-dose corticosteroid remains the standard first-line treatment recommended in most countries.[10]

Compared with cancers in haematology, ITP remains relatively under-researched. The few trials done in ITP have often been funded by pharmaceutical companies, risking publication bias towards high-cost non-generic drugs. For example, many 'cheap', generic drugs commonly prescribed for ITP, such as azathioprine, mycophenolate mofetil (MMF) and dapsone, have never been tested in randomised controlled trials (RCTs) in ITP. In contrast, the more expensive treatments, thrombopoietin receptor agonists (TPO-RAs) and rituximab have been tested in well-designed adequately powered RCTs. The relative rarity (2.9/100 000 person-years), non-cancerous nature and rare impact on survival of ITP have prevented ITP being a priority for research funding

in the past. However, this underestimates the profound adverse impact a diagnosis of ITP and its treatment can have for individual patients, many of whom are young. There is also a costly financial impact for the National Health Service (NHS) from the healthcare resources patients require when their illness is uncontrolled. In addition, the problems faced by patients with ITP mirror those with other autoimmune conditions which as a group are common, affecting 3% of the population. There is an urgent clinical need to address this inequality, improving first-line treatment for ITP through high quality, independently funded research to allow patients with this condition access to improvements in care seen in other illnesses such as cancer or heart disease.

Current popular options for second-line or subsequent treatment include MMF, rituximab, TPO-RA and splenectomy. Splenectomy is an effective treatment (60% long-term remission rates) but irreversible and international guidelines recommend deferring splenectomy for the first 12 months following diagnosis due to the chance of spontaneous remission (risk of unnecessarily removing a healthy organ).[7 11] Surgical operations are not popular with patients and there is increasing awareness of the short-term and long-term complications of splenectomy including infection, bleeding, arterial and venous thrombosis, cancer and relapse.[16] The splenectomy numbers performed in the UK has dramatically reduced over recent years (UK ITP registry data). Rituximab is a monoclonal antibody treatment which targets antibody production by B cells. It is relatively expensive, with disappointing long-term remission rates similar to placebo.[16] TPO-RA stimulate platelet production, are well tolerated and effective in the majority[17] but at significant financial cost, prohibiting widespread use in the UK for early treatment (National Institute for Health and Care Excellence guidance). A small (n=12) non-randomised study using TPO-RA with corticosteroid first line showed efficacy but perhaps less than expected.[18] By contrast, MMF is a widely used second-line agent in the UK due to good efficacy (response rates of 50%–80%), safety and tolerability profile.[19–26] MMF has activity against both autoreactive T and B cells and has also shown efficacy in refractory ITP including steroid resistance suggesting a complimentary mechanism of action.[24] It is less expensive to the NHS than some other second-line options costing approximately £182/year (generic cost) compared with costs for average doses of romiplostim (TPO-RA) at £25 000/year, eltrombopag (TPO-RA) £20 000/year or rituximab at £8000 for a course of 4 infusions (375 mg/m$^2$ each dose) or £1000 (100 mg each dose). However, similar to other second-line therapies, MMF has a relatively slow (up to 2 months) onset of action. In the meantime, patients often receive further steroid (to maintain a 'safe platelet count') and continue to suffer problems associated with their illness (see above). Direct feedback from patients regarding the difficulties they face in the first months following ITP diagnosis has been the primary driving force for this clinical trial. Local (Bristol) and

national patient groups (ITP support association) have been fundamental to the formulation of patient relevant priorities for treatment.

## RATIONALE

The Flight trial is the first UK, NHS coordinated, pharma independent multicentre RCT, testing a 'common sense/practical' new approach using MMF first line instead of second line with the aim of preventing the almost inevitable first relapse when corticosteroids stop. Patients will be randomly allocated to one of two treatment arms, either standard of care (corticosteroid alone) or MMF combined with corticosteroid with the primary outcome of time to treatment failure. By giving patients a stable platelet count sooner, we expect to improve other outcomes such as quality of life and fatigue. By reducing the risk of relapse, patients may also be less likely to receive a second course of corticosteroid with associated side effects. Potential indirect benefits to the NHS include reduced need for rescue treatments, blood tests, hospital attendances and admissions and reduced need for high-cost treatments such as TPO mimetics. However, there will be some patients who will be treated with MMF who may have been successfully treated with corticosteroids alone (10%–30%).[7 9 10] Similar to other immunosuppressives, MMF may slightly increase infection and cancer risk with long-term use (SmPC) In addition, MMF is teratogenic and therefore stringent pregnancy prevention is essential for men and women taking the drug. This puts the trial in equipoise. The trial includes a strategy to reduce and stop MMF at 6 months for patients in complete remission to prevent unnecessary long-term use.

The choice of this open-label design was made in order to allow true patient treatment costs to be calculated for the cost-effectiveness analysis, and to deal with the complexities of placebo controlling a drug that needed titrating at the start and tapering at the end. In addition, over encapsulation was only possible for the lower MMF dose (250 mg) and the resulting capsule was the largest size which would mean most patients taking eight large capsules per day in both arms; something that patients in Bristol thought would put them off taking part in the study. Patients were clear that from their perspective that a straight forward open-label design would be preferable and was easier for a new patient to understand and consent. In addition, the quotes from two separate companies also showed the financial costs of encapsulation to generate a placebo were prohibitively expensive.

This trial proposal has received support and input from clinicians and patients nationally (UK ITP forum and ITP support association). To ensure objective and meaningful outcomes, it will be a multicentre RCT, aiming to recruit 120 patients (expecting 100 full datasets). Patients will be given up to 1 week of corticosteroid prior to randomisation to enable sufficient time to read information, discuss and ask questions with informed consent in an appropriate setting. Patients will receive the usual follow-up according to clinical need and local policy. Laboratory and clinical data will be collected from routine appointments. In addition, patient-oriented outcomes will be recorded at diagnosis, 2, 4, 6 and 12 months using validated patient questionnaires. Patients are also offered consent to additional blood samples for translational research studies (time 0 and 2 months).

### Primary objective

To compare two first-line treatment pathways for ITP, standard corticosteroid only versus corticosteroid combined with MMF and demonstrate which pathway helps patients achieve a stable platelet count sooner, measured as survival free from treatment failure (time from randomisation to treatment failure).

## METHODS AND ANALYSIS

### Trial design

A multicentre, open-label randomised clinical trial of MMF with corticosteroid as first-line treatment for patients with ITP versus the standard care pathway of corticosteroids alone as first-line treatment.

### Eligibility criteria

#### Inclusion criteria

Patients >16 years old with a diagnosis of ITP (primary or secondary), a platelet count $<30\times10^9$/L and a clinical need for first-line treatment.

Patients can be recruited at any time after ITP diagnosis if they are suitable for first-line treatment (ie, not previously or recently treated). Patients can receive up to 1 week of corticosteroid prior to recruitment to allow time to be informed about the trial, with the opportunity to ask questions and for consent to be taken during a routine specialist clinic appointment if preferred.

#### Exclusion criteria

Pregnancy and breast feeding (Women of childbearing potential require a pregnancy test result within 7 days prior to randomisation to rule out unintended pregnancy). Patients with HIV, hepatitis B or C, common variable immunodeficiency. Contraindications to MMF or corticosteroid (see SmPC) including patients with active significant infections, hypersensitivity to MMF, mycophenolic acid or to any of the excipients or active significant infection. Patients not capable of giving informed consent (eg, due to incapacity). Patients (men and women) unwilling to follow contraceptive advice if allocated to MMF treatment arm.

### Study setting

One hundred and twenty patients will be recruited from approximately 40 haematology departments of hospitals (secondary care) across the UK where patients with ITP are treated.

The trial processes will be run by the Centre for Trials Research (CTR), Cardiff University and sponsored by University Hospitals Bristol NHS Foundation Trust.

The flight trial opened for recruitment on 26 October 2017.

## RANDOMISATION

Patients who agree to participate will be randomised to MMF with corticosteroid or corticosteroid alone in a 1:1 ratio using a secure web-based randomisation system based at Cardiff CTR. Randomisation will be stratified by primary or secondary ITP diagnosis. Due to the large number of centres and the small number of patients, it will not be sensible to stratify randomisation by study centre. However, to ensure an even spread of patients across time, randomisation will be blocked using random block sizes of 6 and 8 to retain concealment.

## TREATMENT ARMS

Corticosteroid only pathway figure 1: 1 mg/kg once a day prednisolone for 4 days (maximum of 100 mg), 40 mg once a day for 2 weeks, 20 mg once a day for 2 weeks, 10 mg once a day for 2 weeks, 5 mg once a day for 2 weeks then 5 mg alternate days 2 weeks then stop*. For the duration of steroid, patients will get a PPI or H2 antagonist to protect against gastric bleeding and appropriate bone protection figure 1.

*Dexamethasone 20 mg or 40 mg orally daily for 4 days is an alternative option to prednisolone if deemed clinically more appropriate for individual circumstances.

Corticosteroid +MMF pathway figure 2: 1 mg/kg once a day prednisolone for 4 days (maximum of 100 mg), 40 mg once a day for 2 weeks, 20 mg once a day for 2 weeks, 10 mg once a day for 2 weeks, 5 mg once a day for 2 weeks then 5 mg alternate days 2 weeks then stop*. For the duration of steroid, patients will get a PPI or H2 antagonist to protect against gastric bleeding and appropriate bone protection figure 2.

*Dexamethasone 20 mg or 40 mg orally daily for 4 days is an alternative option to prednisolone if deemed clinically more appropriate for individual circumstances.

From randomisation (alongside steroid), MMF 500 mg two times a day starting dose then increased to 750 mg two times a day after 2 weeks if tolerated (no side effects or laboratory concerns such as neutropenia) and 1 g two times a day after another 2 weeks if tolerated (4 weeks after starting). Earlier dose escalation to MMF 1 g two times a day can be considered if clinically indicated.

After 6 months of MMF therapy, all patients who have remained in complete remission (platelet count $>100\times10^9/L$) will reduce the dose by 250 mg (one capsule) each month. The aim is to continue on the lowest dose that achieves a haemostatic (safe) platelet count (platelet count $>30\times10^9/L$) and to ensure that patients who have gone into a remission do not continue to take the drug indefinitely.

## In both groups

Any steroid commenced prior to randomisation will be deducted from the regimens. Importantly, emergency and rescue treatments will be permitted throughout the study. These include platelet transfusions, tranexamic acid and intravenous Ig. These are known not to impact on the natural history of ITP and it is recognised that they may be important for patient safety. The use of 'rescue treatments' will be recorded on the case report form (CRF).

In addition, some degree of flexibility of corticosteroid dose and duration may be needed for individual patients according to comorbidity, tolerability and other factors.

If treatment failure occurs, choice of second-line treatment will be individualised according to patient's clinical circumstances. Further steroid will be given according to clinical need.

### Primary outcome

Time from randomisation to treatment failure defined as a platelet count $<30\times10^9/L$ and a need to commence second-line treatment. This will include patients who are refractory (platelet count $<30\times10^9/L$ in spite of 2 weeks treatment in the steroid arm or platelet count $<30\times10^9/L$ in spite of 2 months treatment in the steroid +MMF arm) or who initially respond but then relapse (defined clinically as platelet count $<30\times10^9/L$ and need for further therapy). Patients with a clinical need to start second-line treatment early (within 2 weeks for the steroid only arm and within 2 months for the MMF and steroid arm), for example, due to significant bleeding, will also be classed as treatment failures.

### Secondary outcomes

1. Medication side effects, toxicity or other adverse events (including infection episodes).
2. Bleeding events.
   a. Site and type of bleeding.
   b. Treatment required for bleeding.
   c. Whether hospital admission was required.
   d. Whether ITP rescue treatments were needed.
3. Remission rates (platelet count $>30\times10^9/L$ and at least twofold increase from baseline); Complete $100\times10^9/L$, partial $30–100\times10^9/L$.
4. Time to relapse and time to next therapy.
5. Cumulative corticosteroid dose.
6. Need for rescue therapies.
7. Need for splenectomy.
8. Socioeconomic costs.
9. Patient-reported outcomes (quality of life, fatigue, impact of bleeding, care costs).

### Patients follow-up

Patients will be followed up until the end of the trial and for a minimum of 12 months. They will receive the usual follow-up according to clinical need and local policy. Laboratory and clinical data will be collected from routine appointments. In addition, patient-oriented outcomes and additional data will be recorded at diagnosis, 2, 4, 6 and 12 months using validated patient questionnaires. Patients are also offered consent to additional

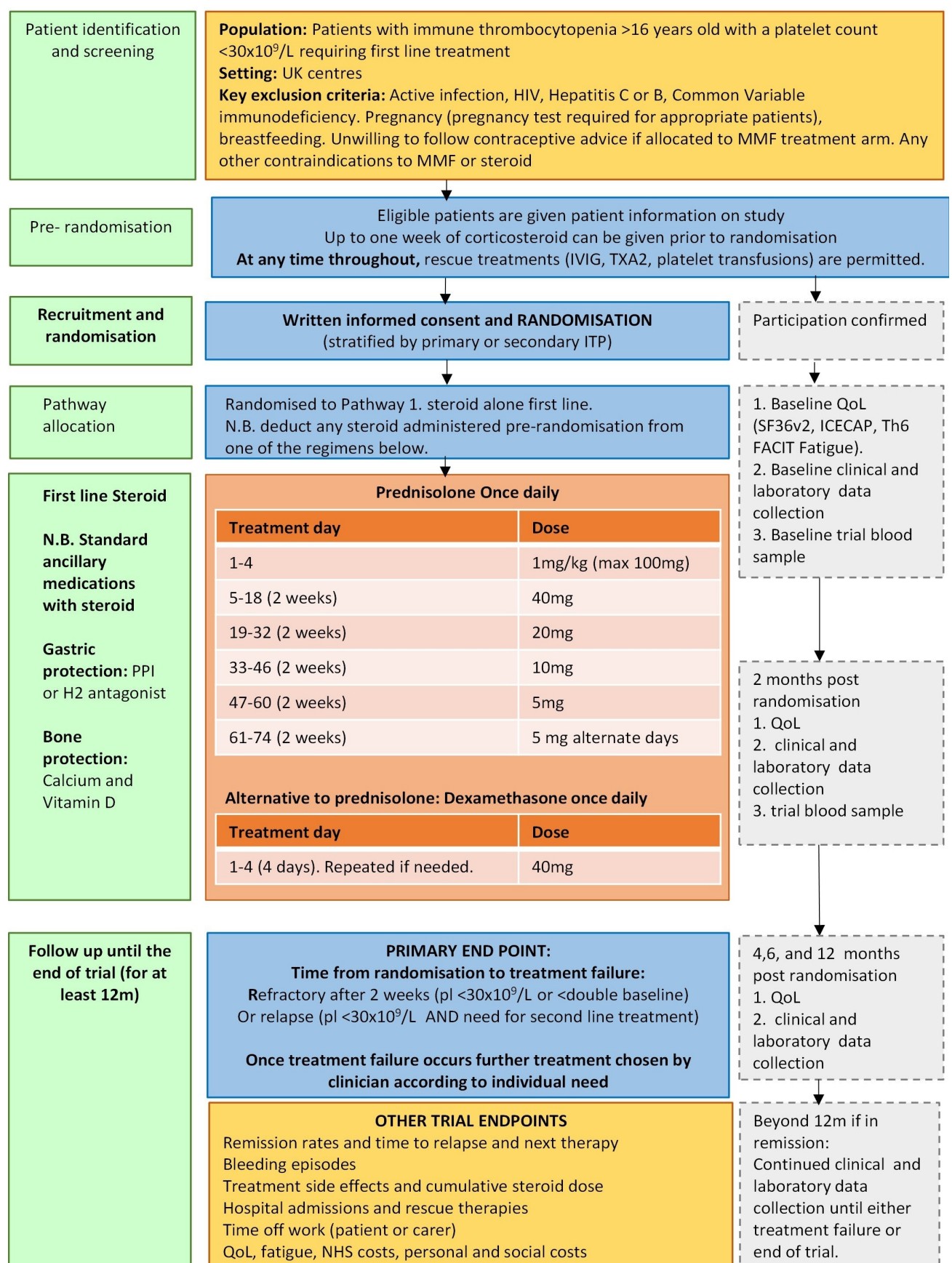

**Figure 1** Flight treatment pathway: corticosteroid only. FACIT, functional assessment of chronic illness therapy; ITP, immune thrombocytopenia; IVIG, intravenous immunoglobulin; MMF, mycophenolate mofetil; NHS, National Health Service; pl, platelet; PPI, proton pump inhibitor; TxA2, Tranexamic Acid; QoL, quality of life.

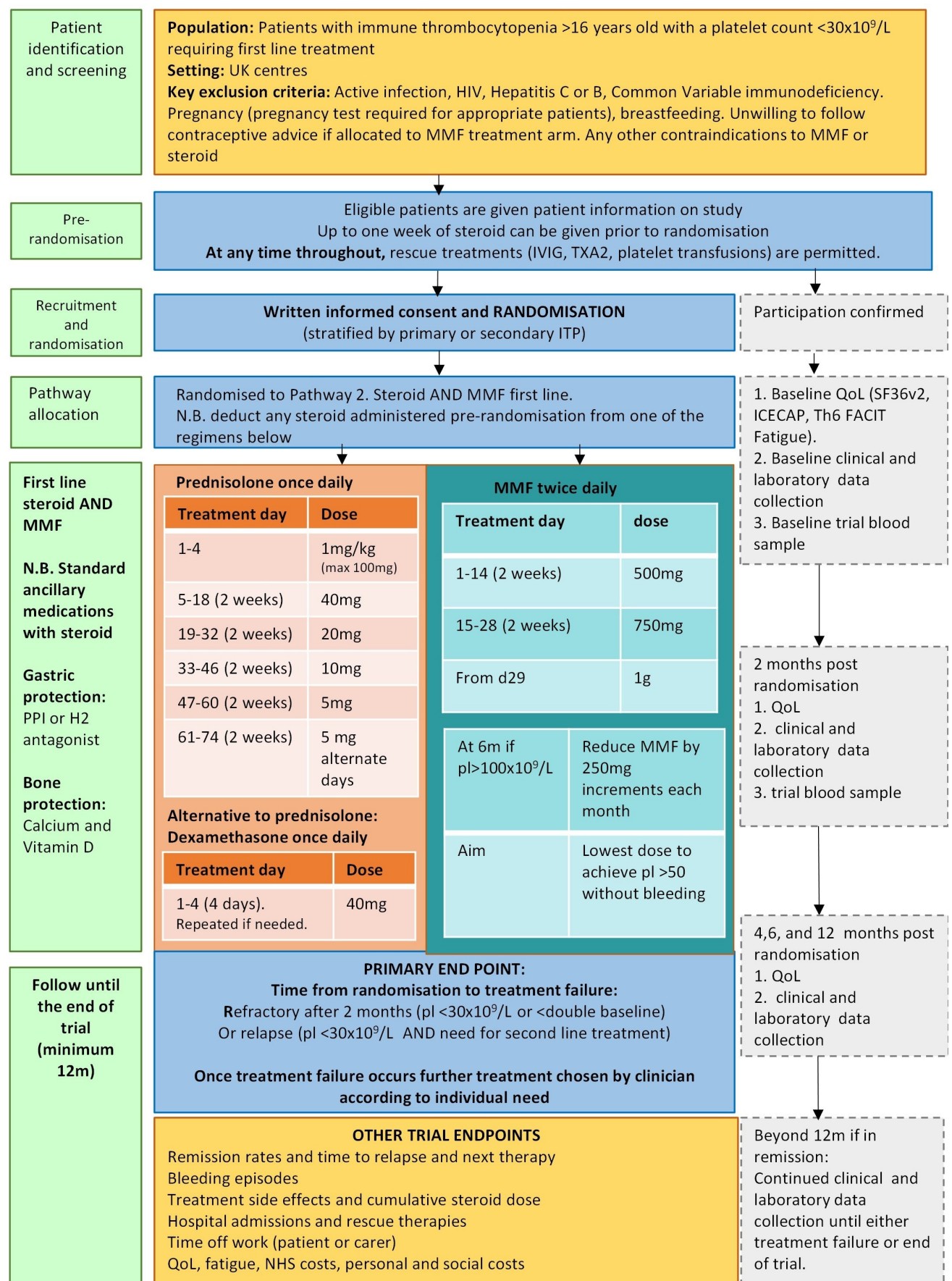

**Figure 2** Flight treatment pathway: corticosteroid and MMF. ITP, immune thrombocytopenia; IVIG, intravenous immunoglobulin; MMF, mycophenolate mofetil; NHS, National Health Service, pl, platelet; QoL, quality of life.

**Table 1** Time schedule of enrolment, interventions, assessments and visits

| Procedures | V0 Screen | V1 Baseline/ randomisation to pathway 1 or 2 | V2 (two months) Follow-up | V3 (four months) Follow-up | V4 (six months) Follow-up | V5 (12 months) Follow-up | V6 12–24 months Data collection from sites |
|---|---|---|---|---|---|---|---|
| Eligibility assessment | x | | | | | | |
| Randomisation | | x | | | | | |
| Informed consent | | x | | | | | |
| Demographics | | x | | | | | |
| Medical history | | x | x | x | x | x | |
| Physical examination | | x | | | | | |
| Vital signs (incl height and weight) | | x | x | x | x | x | |
| Pregnancy test | x | | | | | | |
| Concomitant medications | | x | x | x | x | x | |
| Standard practice bloods (includes blood sugar if applicable) | x | x | x | x | x | x | |
| Hepatitis B, C and HIV serology | x | | | | | | |
| Immunoglobulins (blood) | | x | | | x | x | |
| Extra blood samples (optional) | | x | x | | | | |
| Dispensing of trial drugs | | x* | | | | | |
| Compliance | | | x | | | | |
| QofL FACT-Th6, V.4 | x | x | x | x | x | x | |
| QofL ICECAP V.2—A measure | x | x | x | x | x | x | |
| QofL SF-36V.2— Health Survey | x | x | x | x | x | x | |
| QofL FACIT-F, V.4, pg 3 (fatigue) | x | x | x | x | x | x | |
| QofL Thrombocytopenia costs questionnaire | x | x | x | x | x | x | |
| Data collection from sites on platelet count and treatment | | x | x | x | x | x | x |

*MMF and corticosteroid dispensing frequency can follow standard local practice.
MMF, mycophenolate mofetil; QofL, quality of life.

blood samples for translational research studies (time 0 and 2 months).

## Data collection

Hospital monitoring of platelet levels (FBC) is part of routine care for patients with ITP and these data will be collected and recorded on the CRF without requiring patients to come in for additional samples to be taken. These locally collected samples may be collected monthly (or less often) for patients believed to be in stable remission and weekly at lower or declining platelet levels. We expect this to allow us to calculate the time in remission and time to relapse with reasonable accuracy over the 12–24 months follow-up period. Other clinical and laboratory data needed for the trial endpoints will be collected from the medical and electronic records and recorded on the CRF. In addition, we will also ask the patients to complete questionnaires on fatigue, quality of life and bleeding scores at baseline, 2, 4, 6 and 12 months (table 1).

Patient-reported outcomes will be captured by the following questionnaires:
1. SF36V.2 (Your health & well-being)—quality of life.
2. FACIT-fatigue (V.4)—fatigue.
3. FACT-Th6 (V.4)—bleeding.
4. ICECAP-A V.2—quality of life.
5. Health economic/resource use questionnaire—personal and social costs.

Additional optional research blood samples (requiring separate consent) will be sent at baseline and 2-month follow-up to the Bristol Biobank.

## Data management

Source documents produced for this trial will be kept in the patient's hospital records and source data will be transcribed into trial-specific CRFs at the end of each patient visit. Data recording for this trial will be via a web-based system. This is a secure encrypted system accessed by an institutional password which complies with Data Protection Act standards. The database will be stored and regularly backed up on a Cardiff University Server. The CRFs will be coded with the study number and will not include patients' names and addresses

## Patient and public involvement and engagement

During the trial development, a group of eight patients with ITP discussed the study design, burden of outcome measure completion to patients and the size of a potential

placebo capsule which they reported could put them off getting involved in a trial. They reported that avoidance of relapse, early achievement of a stable platelet count, reduced overall corticosteroid dose and reduced hospital attendances are the most important goals for ITP management from their perspective.

We formed a patient advisory group with some of these patients and representatives from the ITP association that will advise the trial management group (TMG) throughout the study. They have commented on all patient-facing documentation and will be instrumental in disseminating the study findings to patient groups and the public.

## STATISTICS AND DATA ANALYSIS
### Sample size calculation

There are no published clinical data available for MMF use first line in ITP as this is a novel approach. We have analysed local data on MMF used second line in ITP in 12 patients which shows an estimated median survival free from treatment failure of more than 10 months. We have data on 68 who experienced corticosteroids as a first-line treatment showing that 70% of them had experienced a treatment failure by 12 months and that the median survival free from treatment failure was 5.0 months (95% CI 3.2 to 6.8). Data for the 12 patients treated with MMF second-line therapy have shorter follow-up times, with only 5 patients having follow-up beyond 12 months. The Cox proportional hazards regression model demonstrates the 90% CI for the HR to be between 0.13 and 0.59, showing that our decision to power this on an estimate of a hazard of lower than 0.5 is potentially achievable.

Clinically a doubling in the time to remission was thought to be something that the patients would have welcomed. Less than that was not thought to be sufficient grounds for switching this treatment from second line to first line due to the potential for additional toxicity and immune suppression in those who may have remained in remission with corticosteroids alone.

The sample size of 120 (60 per group) with less than 5% loss to follow-up achieves 91.5% power to detect a doubling of the median time to treatment failure from 5 to 10 months if the patients are recruited at a steady rate or 10 per month for 12 months and all followed up until the last patients reaches 12 months follow-up.

### Statistical analysis

The full statistical analysis will be written into a statistical analysis plan available separately. The analysis will produce a Consolidated Standards of Reporting Trials diagram for the reporting of clinical trials.

The baseline characteristics of the two groups will be tabulated but not tested for statistically significant differences between the groups.

The primary analysis is by intention to treat. However, an investigation of compliance with the treatment pathway and compliance with the criteria for changing to a second-line therapy will be carried out prior to the

primary analysis to check the date of the primary event. The primary event is the date at which there was a requirement for second-line therapy. Where the platelet count falls below the level required for this treatment decision, the first date at which either symptoms or a blood test revealing this event will be used. If a clinician decides to use a second-line therapy without a platelet count below the criteria, the date of the treatment decision/new prescription will be taken to represent that event. The results will be expressed as an HR with 95% CI, median time to event if more than 50% have had an event and plotted as Kaplan-Meier curves.

The primary analysis will contain all patients who are randomised for as long as they have been followed up or until their first event in a survival analysis using intention to treat methodology. All patients will be followed up to 12 months. In addition, patients who have not had an event in the first 12 months postrandomisation will be followed until their first event or until the last patient has reached the 12-month point—whichever is the sooner and included in the analysis until that time accordingly. Sensitivity analyses will include landmark analysis or shifting the time line to classify all treatment failures before 2 months as at 2 months in order to prevent potential biases caused by different definitions of treatment failure time frames between the two groups.

Analysis of other outcomes will use as full a data set as possible and focus on the 12-month data point or area under the curve as appropriate and detailed in the analysis plan.

No interim analyses of the main endpoint will be supplied to the independent data monitoring committee (DMC) due to the short time frame (12 months recruitment) in which all patients will be recruited by the time the first patient has completed follow-up. Serious adverse event (SAE) rates will be reported on a monthly basis to the TMG and the DMC. The DMC could advise the chairman of the trial steering committee and chief investigator if these provide proof beyond reasonable doubt that it would be unethical to continue with the trial.

### Pharmacovigilance

The collection and reporting of all adverse events is in accordance with the Medicines for Human Use Clinical Trials Regulations 2004 and its subsequent amendments and follows the standard operating procedures of the trials unit, Cardiff University CTR.

Seriousness and causality are assessed by participating sites and further review of expectedness (based on the reference safety information) is conducted centrally on behalf of the Sponsor. Events are defined as SAEs, serious adverse reactions (SARs) or suspected unexpected SAR in line with regulatory definitions on the basis of these assessments.

SAEs are reported throughout the treatment period up to 6 weeks after cessation of last dose of MMF. SARs should continue to be reported until the end of follow-up. All deaths and overdoses are reported to the sponsor as an SAE and reviewed in line with other events.

MMF in this trial has a genotoxic and teratogenic potential and therefore pregnancy is contraindicated. Participants who are female of childbearing potential or male with female partners of equal potential are required to use contraception as indicated in the protocol. Pregnancy or the pregnancy of a partner occurring while participating in the trial is not considered an SAE, however, a congenital anomaly or birth defect is. Pregnancy is reported to sponsor and followed up to outcome.

All safety events are reviewed by the TMG on an ongoing basis and reported to the trial steering committee for oversight. Overall assessment of the safety profile of both arms will be included in final reporting and publication.

**Author affiliations**
[1]Centre for Trials Research, Cardiff University, Cardiff, Wales, UK
[2]Research and Design Service, South West, University of Bristol, Bristol, UK
[3]Research & Innovation, University Hospitals Bristol NHS Foundation Trust, Bristol, UK
[4]Cellular and Molecular Medicine, University of Bristol, Bristol, UK
[5]Department of Haematology, University Hospitals Bristol NHS Foundation Trust, Bristol, UK
[6]UCL-Institute of Ophthalmology, London, UK
[7]Department of Haematology, Glasgow Royal Infirmary, Glasgow, UK
[8]Department of Haematology, Imperial College London and Hammersmith Hospital, London, UK
[9]Department of Haematology, Leeds Teaching Hospitals NHS Trust, Leeds, UK

**Acknowledgements** We would like to thank the members of the UK ITP forum who have been invaluable providing feedback on the trial design and recruitment to the trial.

**Contributors** CAB is chief investigator for the Flight trial and was responsible for writing the protocol with clinical input from NC, QH and CB. RG is the trial statistician who has contributed to the trial design and writing of protocol. JI contributed to trial design and leads the patient advisory group input. RK is the trial health economist and provided input to the trial design and protocol. JP and IT (Cardiff CTR) and KW (sponsor representative) have also provided contribution to writing the protocol. AM and AD have provided mentorship to the chief investigator.

**Funding** The FLIGHT trial is independent research funded by the National Institute for Health Research (NIHR) under its Research for Patient Benefit (RfPB) Programme (Grant Reference Number PB-PG-0815- 20016).

**Disclaimer** The views expressed are those of the authors and not necessarily those of the NHS, the NIHR or the Department of Health.

**Competing interests** None declared.

**Patient consent** Not required.

**Ethics approval** Ethical approval from NRES Committee South West (IRAS number 225959).

**Provenance and peer review** Not commissioned; externally peer reviewed.

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
