## [Reviewer comments · BMJ Open]

ARTICLE DETAILS

TITLE (PROVISIONAL)	Trial Protocol: A multicentre randomised trial of First Line treatment pathways for newly diagnosed Immune Thrombocytopenia: Standard steroid treatment versus combined steroid and mycophenolate. The FLIGHT trial
AUTHORS	Pell, Julie; Greenwood, Rosemary; Ingram, Jenny; Wale, Katharine; Thomas, Ian; Kandiyali, Rebecca; Mumford, Andrew; Dick, Andrew; Bagot, Catherine; Cooper, Nichola; Hill, Quentin; Bradbury, Charlotte

VERSION 1 – REVIEW

REVIEWER	David Gómez-Almaguer Universidad Autónoma de Nuevo León , Monterrey, México
REVIEW RETURNED	19-Jun-2018

GENERAL COMMENTS	This is an interesting proposal since mycophenolate has not been tested as a first line treatment in ITP. The protocol is well written and the idea is innovative in this field. However, there are some observations I want to remark. *I consider it is important to define the ITP diagnostic criteria according to international guidelines (newly ?, persistent ? etc) *It is included, as a secondary outcome, the report of bleeding events. It would be appropriate to describe these findings according to the following article "Standardization of bleeding assessment in immune thrombocytopenia: report from the International Working Group". Blood 2013 121:2596-2606. *You must mention the starting date (day one after diagnosis ?) of the protocol and treatment. *You must explain in more detail the randomization system that will be used *It is important to specify if you are going to include patients with primary ITP or secondary ITP or both.
--

REVIEWER	Cindy Neunert Columbia University Medical Center, New York, NY, USA
REVIEW RETURNED	25-Jun-2018

GENERAL COMMENTS	June 22, 2018 Re: bmjopen-2018-024427 Title: "A multicentre randomised trial of Frist Line treatment pathways for newly diagnosed immune thrombocytopenia: Standard steroid treatment versus combined steroid and mycophenolate. The FLIGHT trial"
---

	The authors present a study outline for a trial comparing standard corticosteroid therapy to corticosteroid therapy augmented with mycophenolate (MMF). This is a novel study and answers an important question related to improving outcomes for adults with immune thrombocytopenia (ITP). The trial is well outlined and my specific comments can be found below: Major Comments:  1. The estimated doubling in time from 5-10 months would be nice to have in abstract. 2. Some places in the background/introduction would benefit from a reference. Such as the mention of the few trials that have been done and supported by industry (page 4 line 23) 3. For the refractory patients it would seem that this will naturally influence the time to treatment failure if one group is allowed only 2 weeks of a platelet count <30K before being calling refractory and the other 2 months. How will the authors handle this in the analysis? 4. Please make sure listing of secondary outcomes is consistent throughout the document. There are some differences between the abstract, page 7 and page 9. 5. How many cycles of dexamethasone will be allowed before considering a patient refractory and/or a treatment failure? Minor Comments:  1. The statement about “any steroid use...” on page 8 line 37 and page 9 line 12 is redundant as it is stated for both groups on page 9 line 17. 2. I am not sure that it needs to be expressed that patients are both males and females as gender is not an exclusion criteria. 3. The use of steroids for > 7 days prior to enrollment and/or the duration of ITP allowed at enrollment should be stated somewhere in the inclusion and exclusion criteria.
--	---

REVIEWER	Waleed Ghanima Departments of Research and Medicine, Østfold Hospital and Department of Haematology, Institute of Clinical Medicine, University of Oslo, Norway.
REVIEW RETURNED	07-Jul-2018

GENERAL COMMENTS	This manuscript describes a study protocol for a multicentre randomised trial comparing two first line treatment pathways for newly diagnosed ITP, involving an open label treatment with standard steroid versus combined steroid and mycophenolate. MMF is frequently used as a second line treatment for ITP in the UK, and generally as third line in some other countries, despite the lack of solid evidence from well-designed RCTs. There is therefore an unmet need for RCT to evaluate the effect and safety of MMF in ITP. This trial brings therefore MMF one step forward and tests the effect of MMF in combination with steroids as upfront treatment. Following are my comments Major comments 1- The authors state that “the first line treatment for ITP is unsatisfactory but it has been unchallenged for decades.” Dexamethasone in combination with Rituximab has shown
--

superiority over dexamethasone alone as upfront treatment. This combination is given as a standard first line treatment at least in Denmark. Therefore, this statement should be amended. However, despite favorable results, the combination has not gained wide acceptance, mainly because it is well-known that 30% achieve remission by steroid alone. I am therefore worried that the proposed regimen of steroids+MMF will have the same destiny!

2- The authors admit that there will be some patients who will be treated with MMF who may have been successfully treated with corticosteroids alone (10-30%), however, they provide no rationale for why they did not provide the combination to patients who fails at least a short trial of steroids e.g. 4 weeks or one course of dexamethasone? Please discuss.

3- My main concern is the difference in time to the introduction of new treatment after treatment failure in the two arm. In my opinion you are introducing a systematic bias in favor of MMF. This could have been prevented by a placebo controlled design. Please discuss.

4- The authors state that “The choice of this open label design was made in order to allow true patient treatment costs to be calculated for the cost effectiveness analysis, and to deal with the complexities of placebo.” While I acknowledge the simplicity of open design, I don’t think that a placebo controlled study would have impacted so negatively on cost effectiveness analysis. Please revise this statement. I believe as I indicated under point 3, a placebo controlled design would have prevented a possible bias resulting from open treatment allocation. Moreover, there will always be a tendency to delay the introduction of a new treatment (a requirement for the primary outcome) in the MMF arm as opposite to the steroid arm.

5- Since you are introducing a new therapy with the potential of having serious toxicity, safety should have been considered either as a second primary endpoint, or main secondary endpoint, and not as the last of secondary outcome! Please consider this point.

6- Please include a section on pharmacovigilance.

7- Please include a section on “patients’ follow up” after “outcomes” to describe how the patients will be followed-up and for how long. This information comes first in “statistics” section.

8- Please provide NCT number. To my knowledge this study is already recruiting, however, study status on clinicaltrials.gov is “Not recruiting.” Please comment.

9- There is no mention on how bleeding will be captured and graded. This information is crucial to a trial in ITP. Beside, in my opinion, bleeding should not be considered as an adverse event in an ITP trial as stated in the manuscript. Please clarify and correct.

Minor comments:

1- I do not understand the 1st secondary endpoint “data on time to treatment failure as measured objectively by a fall in platelet count to $<30 \times 10^9/L$ and need for new treatment.” What is the difference

	between this and the primary endpoint? Please clarify. Besides, what is actually meant by “data”? I suggest that you delete this word. 2- Eligibility criteria. Not clear to me if only primary ITP are allowed and whether relapsed ITP is allowed. Also state early in the manuscript that that she study will include primary and secondary ITP. 3- P 3 R 56: (Newland A et al, poster BSH 20159) should be added to reference list. 4- P 6 R 19: is stated that the cost of rituximab is £8000 for a course of 4 infusions. Rituximab is often given in much lower doses in the UK. Therefore, a range of cost should be provided. 5- MMF 500mg bd starting dose then increased to 750mg bd after 2 weeks if tolerated - what are the signs of intolerance? 6- P 3 R 39: ITP does not always present with bleeding. Please correct 7- P 8 R 52: “to ensure that patients who have gone into a spontaneous remission do not continue to take the drug indefinitely” – delete spontaneous as patients are on medications and therefore the remission is not spontaneous.
--	---

VERSION 1 – AUTHOR RESPONSE

Reviewer(s)' Comments to Author:

Reviewer: 1

Reviewer Name: David Gómez-Almaguer

Institution and Country: Universidad Autónoma de Nuevo León , Monterrey, México

Please state any competing interests or state 'None declared': None declared

Please leave your comments for the authors below

This is an interesting proposal since mycophenolate has not been tested as a first line treatment in ITP. The protocol is well written and the idea is innovative in this field. However, there are some observations I want to remark.

*I consider it is important to define the ITP diagnostic criteria according to international guidelines (newly ?, persistent ? etc)

This has been added to the introduction.

*It is included, as a secondary outcome, the report of bleeding events. It would be appropriate to describe these findings according to the following article "Standardization of bleeding assessment in immune thrombocytopenia: report from the International Working Group". Blood 2013 121:2596-2606. Thank you for this comment. We have now clarified the data we are collecting on bleeding episodes which includes details on site and management with specific required tick boxes on if bleeding is mucocutaneous only or resulted in hospital admission or ITP rescue treatments. We would expect this

data to be sufficiently detailed to sub classify bleeds according to standard criteria after the trial if needed. Following early discussions with clinicians across the UK, a pragmatic approach to collection of bleeding events was chosen for this trial rather than using a standardized tool. Of note, we are also using a patient reported outcome measure of bleeding. As stated above, the trial has already opened for recruitment and therefore unfortunately it is not possible to amend the CRF at this stage.

*You must mention the starting date (day one after diagnosis ?) of the protocol and treatment. We have clarified this in the inclusion criteria.

*You must explain in more detail the randomization system that will be used
We are using a secure web based randomization system stratified by primary versus secondary ITP and using variable block sizes to ensure both concealment and an even recruitment of both study arms throughout the recruitment time frame.

*It is important to specify if you are going to include patients with primary ITP or secondary ITP or both. This is now clarified in the inclusion criteria and the study setting. We are recruiting patients with primary and secondary ITP and stratifying by this variable.

Reviewer: 2

Reviewer Name: Cindy Neunert

Institution and Country: Columbia University Medical Center, New York, NY, USA

Please state any competing interests or state 'None declared': None

Please leave your comments for the authors below

June 22, 2018

Re: bmjopen-2018-024427

Title: "A multicentre randomised trial of Frist Line treatment pathways for newly diagnosed immune thrombocytopenia: Standard steroid treatment versus combined steroid and mycophenolate. The FLIGHT trial"

The authors present a study outline for a trial comparing standard corticosteroid therapy to corticosteroid therapy augmented with mycophenolate (MMF). This is a novel study and answers an important question related to improving outcomes for adults with immune thrombocytopenia (ITP). The trial is well outlined and my specific comments can be found below:

Major Comments:

1. The estimated doubling in time from 5-10 months would be nice to have in abstract.

This is now included.

2. Some places in the background/introduction would benefit from a reference. Such as the mention of the few trials that have been done and supported by industry (page 4 line 23)

This point has been clarified in the introduction.

3. For the refractory patients it would seem that this will naturally influence the time to treatment failure if one group is allowed only 2 weeks of a platelet count <30K before being calling refractory and the other 2 months. How will the authors handle this in the analysis?

This is correct and recognized as a potential but unavoidable bias due to the slower mechanism of action of MMF. Time to treatment failure will be influenced by the length of time it takes for the clinical staff to be sure that the treatment is not working. This trial design was discussed within the UK ITP forum and clinically it was decided that in real life 2 months would be needed before deciding MMF was not efficacious but in contrast, 2 weeks was a reasonable time frame for corticosteroids.

However, this is not the only reason for treatment failure and patients may also become treatment failures if they have an adverse reaction to the treatment or if they have initially responded and then relapse. Each of these possibilities has a different hypothesized time to treatment failure and the clinical question we are asking is whether the need to step up to the next therapy will be greater in one arm of this trial compared with the other arm. Because the proportions and timings of these three different forms of treatment failure are expected to be different, we will examine the different shaped survival curves in the first instance by plotting the Kaplan Meier curves as stated in the paper. Of note, the proportion of patients expected to be refractory is much smaller than those that reach the treatment failure primary end point due to relapse as the majority of patients (approx. 80%) do initially respond to corticosteroid alone and high relapse rates are the greater clinical problem.

4. Please make sure listing of secondary outcomes is consistent throughout the document. There are some differences between the abstract, page 7 and page 9. The trial objectives were on page 7 and the trial outcomes were on page 9 with minor differences. We have ensured the outcomes match the abstract and NCT database and removed the secondary objectives to keep the word count down as these seemed unnecessary and repetitive.

5. How many cycles of dexamethasone will be allowed before considering a patient refractory and/or a treatment failure?

The majority of ITP treating clinicians in the UK still use prednisolone but some centers were keen to have dexamethasone as an option. We would expect most patients to receive 1 or 2 cycles of dexamethasone but in some cases up to 4 cycles may be needed. Following discussions with our trial steering group we have intentionally allowed some flexibility of steroids (dexamethasone and prednisolone) in both arms so that clinicians felt comfortable to recruit and are specifically collecting data on what corticosteroids patients actually receive (dose and duration) which is included as a secondary outcome. Our patient advisory group has reported that a reduction in the overall amount of steroid is an important endpoint from their perspective.

Minor Comments:

1. The statement about “any steroid use...” on page 8 line 37 and page 9 line 12 is redundant as it is stated for both groups on page 9 line 17.

We have amended this. Thank you.

2. I am not sure that it needs to be expressed that patients are both males and females as gender is not an exclusion criteria.

These have been deleted. Thank you.

3. The use of steroids for > 7 days prior to enrollment and/or the duration of ITP allowed at enrollment should be stated somewhere in the inclusion and exclusion criteria.

A clarifying paragraph has been added to the eligibility criteria:

Patients can be recruited at any time after ITP diagnosis if they are suitable for first line treatment (i.e. Not previously or recently treated). Patients can receive up to 1 week of corticosteroid prior to recruitment to allow time to be informed about the trial, with the opportunity to ask questions and for consent to be taken during a routine specialist clinic appointment if preferred.

Reviewer: 3

Reviewer Name: Waleed Ghanima

Institution and Country: Departments of Research and Medicine, Østfold Hospital and Department of Haematology, Institute of Clinical Medicine, University of Oslo, Norway.

Please state any competing interests or state 'None declared': None

Please leave your comments for the authors below

This manuscript describes a study protocol for a multicentre randomised trial comparing two first line treatment pathways for newly diagnosed ITP, involving an open label treatment with standard steroid versus combined steroid and mycophenolate.

MMF is frequently used as a second line treatment for ITP in the UK, and generally as third line in some other countries, despite the lack of solid evidence from well-designed RCTs. There is therefore an unmet need for RCT to evaluate the effect and safety of MMF in ITP. This trial brings therefore MMF one step forward and tests the effect of MMF in combination with steroids as upfront treatment.

Following are my comments

Major comments

1- The authors state that “the first line treatment for ITP is unsatisfactory but it has been unchallenged for decades.” Dexamethasone in combination with Rituximab has shown superiority over dexamethasone alone as upfront treatment. This combination is given as a standard first line treatment at least in Denmark. Therefore, this statement should be amended. However, despite favorable results, the combination has not gained wide acceptance, mainly because it is well-known that 30% achieve remission by steroid alone. I am therefore worried that the proposed regimen of steroids+MMF will have the same destiny!

We have reworded this to “unchanged” rather than “unchallenged” (in most parts of the world high dose corticosteroid remains the recommended first line treatment). The trial for first line rituximab with dexamethasone treatment showed superiority at 6 months but with increased toxicity and meta-analysis for 2nd line rituximab treatment shows no difference to placebo for long term follow up. We are aware that some subgroups (e.g. young women) may have sustained long term efficacy with rituximab.

In the UK we benefit from a committee (NICE) who routinely produce guidance on treatment options for use in the UK based on both effectiveness and cost effectiveness to the NHS. We expect that if this study shows both effectiveness and cost effectiveness for a specific clinical pathway, that NICE will issue guidance likely to alter treatment options for NHS patients. We are not aware of a cost effectiveness analysis carried out with the Rituximab data and that without that information NICE are unlikely to recommend a change in first line therapy.

2- The authors admit that there will be some patients who will be treated with MMF who may have been successfully treated with corticosteroids alone (10-30%), however, they provide no rationale for why they did not provide the combination to patients who fails at least a short trial of steroids e.g. 4 weeks or one course of dexamethasone? Please discuss.

Patients who fail corticosteroids alone have met their primary endpoint and can have any treatment decided by their clinician which in the UK is often MMF +/- further corticosteroid (TPO RA, rituximab and azathioprine are common alternatives). Beyond the primary endpoint patients continue to be

followed up for data collection but treatment is not dictated by the protocol (beyond the primary endpoint the trial is predominantly observational)

3- My main concern is the difference in time to the introduction of new treatment after treatment failure in the two arm. In my opinion you are introducing a systematic bias in favor of MMF. This could have been prevented by a placebo controlled design. Please discuss.

As mentioned in the introduction, we researched a placebo controlled trial but this was challenging with MMF due to the largest capsule size needed and the only tablet that could be encapsulated was the 250mg one which would require 8 tablets per day for the 1g bd dose. When discussed with our patient advisory group they thought this would put them off being recruited to the study and they preferred the open label design. In addition, the quotes from 2 separate companies were in the region of £100,000 which would not be affordable within the NIHR RFPB funding stream.

We have included the open label design as a limitation in the list of “strengths and limitations”, however it does afford us the luxury of properly quantifying the true costs associated with the different treatment pathways and gives us a much better estimate of the level of compliance and effect size we might see if either of these clinical pathways were to be used as standard of care. Any systematic bias in favour of MMF that might be seen in this study is also likely to be present in routine clinical care.

Please see above response on the different time frames in the 2 arms for the definition of refractory patients (reviewer 2 point 3).

4- The authors state that “The choice of this open label design was made in order to allow true patient treatment costs to be calculated for the cost effectiveness analysis, and to deal with the complexities of placebo.” While I acknowledge the simplicity of open design, I don’t think that a placebo controlled study would have impacted so negatively on cost effectiveness analysis. Please revise this statement. I believe as I indicated under point 3, a placebo controlled design would have prevented a possible bias resulting from open treatment allocation. Moreover, there will always be a tendency to delay the introduction of a new treatment (a requirement for the primary outcome) in the MMF arm as opposite to the steroid arm.

Within the NHS the placebo effect of any treatment must be included in the cost effectiveness analysis as this represents the true benefit of prescribing this treatment to the NHS and to the patient. Although this might mean that a placebo could be cost effective in its own right, we don’t have a way to prescribe placebos and so including a placebo arm in a study will only prove efficacy and not effectiveness. We wish to demonstrate effectiveness as efficacy has already been demonstrated for MMF.

5- Since you are introducing a new therapy with the potential of having serious toxicity, safety should have been considered either as a second primary endpoint, or main secondary endpoint, and not as the last of secondary outcome! Please consider this point.

This is no longer the last secondary outcome. MMF has been used as standard second line therapy for ITP in many centres in the UK and the published evidence (albeit not from randomized studies) supports that when MMF is used for this indication it is well tolerated.

6- Please include a section on pharmacovigilance.
This is now added. Thank you.

7- Please include a section on “patients’ follow up” after “outcomes” to describe how the patients will be followed-up and for how long. This information comes first in “statistics” section.

This is now added. Thank you.

8- Please provide NCT number. To my knowledge this study is already recruiting, however, study status on clinicaltrials.gov is “Not recruiting.” Please comment.

This is now added. Thank you.

9- There is no mention on how bleeding will be captured and graded. This information is crucial to a trial in ITP. Beside, in my opinion, bleeding should not be considered as an adverse event in an ITP trial as stated in the manuscript. Please clarify and correct.

Bleeding data is collected in detail and separate to adverse event data. The data we are collecting on bleeding events is now clarified.

Minor comments:

1- I do not understand the 1st secondary endpoint “data on time to treatment failure as measured objectively by a fall in platelet count to $<30 \times 10^9/L$ and need for new treatment.” What is the difference between this and the primary endpoint? Please clarify. Besides, what is actually meant by “data”? I suggest that you delete this word.

We agree. As the secondary objectives were largely repeating the secondary endpoints we have removed these all together from the manuscript to avoid repetition.

2- Eligibility criteria. Not clear to me if only primary ITP are allowed and whether relapsed ITP is allowed. Also state early in the manuscript that that she study will include primary and secondary ITP. This is now clearer in the eligibility criteria. Thank you for bringing this to our attention.

3- P 3 R 56: (Newland A et al, poster BSH 20159) should be added to reference list.

This is now added. Thank you.

4- P 6 R 19: is stated that the cost of rituximab is £8000 for a course of 4 infusions. Rituximab is often given in much lower doses in the UK. Therefore, a range of cost should be provided.

This is now included

5- MMF 500mg bd starting dose then increased to 750mg bd after 2 weeks if tolerated - what are the signs of intolerance?

Dose is increased if there are no laboratory side effects (e.g. neutropenia or liver dysfunction) or clinical side effects (e.g. diarrhoea or infection). This is now added. Thank you.

6- P 3 R 39: ITP does not always present with bleeding. Please correct
Have corrected to “may” present with bleeding and bruising. Thank you.

7- P 8 R 52: “to ensure that patients who have gone into a spontaneous remission do not continue to take the drug indefinitely” – delete spontaneous as patients are on medications and therefore the remission is not spontaneous.

This is now deleted. Thank you.

VERSION 2 – REVIEW

REVIEWER	David Gomez-Almaguer Univ Autonoma Nuevo Leon, Hematologia
REVIEW RETURNED	22-Aug-2018

GENERAL COMMENTS	I think that now is acceptable, considering that is a protocol.
---

REVIEWER	Cindy Neunert Columbia University Medical Center
REVIEW RETURNED	27-Aug-2018

GENERAL COMMENTS	The authors have adequately addressed my concerns and the overall study design is easy to follow with the current revisions. I have no further comments for the authors.
--

REVIEWER	Waleed Ghanima Institution and Country: Departments of Research and Medicine, Østfold Hospital and Department of Haematology, Institute of Clinical Medicine, University of Oslo, Norway.
REVIEW RETURNED	18-Aug-2018

GENERAL COMMENTS	The authors have adequately responded to almost all my questions and comments. However, I am still concerned about the bias caused by the difference in assessing time to treatment failure in the 2 groups, which will naturally influence the time to treatment failure – an issue that reviewer 2 has also raised. I have no problem in understanding the rationale and the recommendation of the UK ITP forum, however, this is a scientific trial where the methodology should be correct. Assuming that all patients in both groups will fail therapy due to lack of response, then you will still get a significant difference in effect, or am I wrong? One possible solution is to use a landmark analysis where the primary endpoint is changed from time to treatment failure to response rates at 2 or 3 or 6 months. The other is to keep the same design but use an equal time point in both groups from which the clock will start e.g form 2 months. Of course, you have to discuss this with your statistician.
--

VERSION 2 – AUTHOR RESPONSE

The authors have adequately responded to almost all my questions and comments. However, I am still concerned about the bias caused by the difference in assessing time to treatment failure in the 2 groups, which will naturally influence the time to treatment failure – an issue that reviewer 2 has also raised. I have no problem in understanding the rationale and the recommendation of the UK ITP forum, however, this is a scientific trial where the methodology should be correct. Assuming that all patients in both groups will fail therapy due to lack of response, then you will still get a significant difference in effect, or am I wrong?

One possible solution is to use a landmark analysis where the primary endpoint is changed from time to treatment failure to response rates at 2 or 3 or 6 months. The other is to keep the same design but use an equal time point in both groups from which the clock will start e.g form 2 months. Of course, you have to discuss this with your statistician.

We are very grateful for this helpful suggestion and will perform sensitivity analysis as suggested to adjust for this potential bias (the manuscript text has been altered to reflect this). We have also clarified in the methods that patients with a clinical need for early second line treatment within the time window of 2 weeks for steroid or 2 months for MMF and steroid are also classed as treatment failures (e.g. due to severe bleeding)